# ADA-K ROUTING: BOOSTING THE EFFICIENCY OF MOE-BASED LLMS

**Tongtian Yue[1,2], Longteng Guo[1,2], Jie Cheng[1,2], Xuange Gao[1,2], Hua Huang[3], Jing Liu[1,2]***

[1]Institute of Automation, Chinese Academy of Sciences
[2]School of Artificial Intelligence, University of Chinese Academy of Sciences
[3]School of Artificial Intelligence, Beijing Normal University

## ABSTRACT

In the era of Large Language Models (LLMs), Mixture-of-Experts (MoE) architectures offer a promising approach to managing computational costs while scaling up model parameters. Conventional MoE-based LLMs typically employ static `Top-K` routing, which activates a fixed and equal number of experts for each token regardless of their significance within the context. In this paper, we propose a novel `Ada-K` routing strategy that dynamically adjusts the number of activated experts for each token, thereby improving the balance between computational efficiency and model performance. Specifically, our strategy incorporates learnable and lightweight allocator modules that decide customized expert resource allocation tailored to the contextual needs for each token. These allocators are designed to be fully pluggable, making it broadly applicable across all mainstream MoE-based LLMs. We leverage the Proximal Policy Optimization (PPO) algorithm to facilitate an end-to-end learning process for this non-differentiable decision-making framework. Extensive evaluations on four popular baseline models demonstrate that our `Ada-K` routing method significantly outperforms conventional `Top-K` routing. Compared to `Top-K`, our method achieves over 25% reduction in FLOPs and more than 20% inference speedup while still improving performance across various benchmarks. Moreover, the training of `Ada-K` is highly efficient. Even for Mixtral-8x22B, a MoE-based LLM with more than 140B parameters, the training time is limited to 8 hours. Detailed analysis shows that harder tasks, middle layers, and content words tend to activate more experts, providing valuable insights for future adaptive MoE system designs. The code and checkpoints will be released at `https://github.com/ivattyue/Ada-K`.

## 1 INTRODUCTION

Over the past few years, the rapid development of Large Language Models (LLMs) (Brown et al., 2020b; Raffel et al., 2020; Touvron et al., 2023a; Chiang et al., 2023) has marked a significant leap towards Artificial General Intelligence (AGI). Generally, increasing the number of parameters in an LLM enhances its performance across diverse tasks, demonstrating emergent capabilities (Kaplan et al., 2020; Brown et al., 2020a). However, this improvement comes with substantial computational costs for both training and inference, posing barriers to the broad application and efficiency.

In response to these challenges, the Mixture-of-Experts (MoE) (Jacobs et al., 1991; Eigen et al., 2013) architecture has gained popularity as a scalable solution that balances parameter increase with computational cost. MoE implementations in Transformers (Vaswani et al., 2017) have shown that significant model scaling can be achieved without a proportional rise in computational burden, thus maintaining efficient performance. These successes highlight the promising potential of MoE-based LLMs (Jiang et al., 2024; Team; Dai et al., 2024).

The core of the MoE architecture comprises a set of expert networks governed by a routing strategy. This routing strategy, executed via a learnable router (Fedus et al., 2022a; Du et al., 2022a), selectively assigns each token to a limited number of experts. This sparse expert selection is pivotal for MoE efficiency. The most common routing strategy is `Top-K` routing (Shazeer et al., 2017b). It selects the best-suited experts for each input based on preliminary probability calculations, activating the top $k$ experts. Although recent studies (Zoph et al., 2022; Lewis et al., 2021a) have introduced adjustments

---

*Corresponding author.

to ensure more balanced activation across experts, a major limitation remains: the fixed activation numbers does not account for the varying importances of different tokens.

The importances of tokens can vary significantly, potentially influenced by factors such as the different demands of tasks (Rogers et al., 2021), the inherent characteristics of words (Schick & Schütze, 2020), and the contexts in which they are used (Guu et al., 2020). This oversight of token differences can lead to resource inefficiencies. Simpler tokens with minimal semantic significance may receive more processing power than needed, leading to inefficiencies. Conversely, more complex tokens—those representing critical information or requiring advanced logical reasoning—might not get adequate attention. This misallocation of expert resources not only suboptimizes performance but also hinders further improvements in computation efficiency.

To address the limitations of traditional `Top-K` routing in MoE models, we propose a novel, learnable `Ada-K` routing strategy that adapts expert allocation based on each token's inherent importance and difficulty. Specifically, this strategy introduces a pluggable, lightweight allocator, easily integrable into existing MoE-based LLMs. The allocator dynamically determines the optimal number of experts for each token by sampling from the probability distribution it outputs over the possible expert counts. However, this sample operation is inherently non-differentiable. Hence, we employ the Proximal Policy Optimization (PPO) algorithm (Stiennon et al., 2020; Nakano et al., 2021) to optimize the allocators end-to-end, towards ideal balance between model performance and efficiency.

We validate the effectiveness of our proposed `Ada-K` routing strategy by integrating it into four popular MoE-based LLMs and assessing their performance across multiple benchmarks. Compared to baseline models utilizing `Top-K` routing, our `Ada-K` routing consistently reduces the number of activated experts by 30% to 40%, while simultaneously enhancing overall performance. This reduction in expert activation directly translates to computational gains, achieving more than a 25% reduction in FLOPs and a 20% speedup in inference time. Notably, this favorable balance is attained with minimal additional training, requiring the tuning of only approximately 2M parameters, with training times under 8 hours for all baseline models.

Furthermore, we conduct a detailed analysis of the `Ada-K` routing strategy at the task, layer, and token levels. Our findings indicate that harder tasks, middle layers, and content words tend to activate more experts, demonstrating the strategy's ability to allocate computational resources efficiently based on the importance of input tokens. These conclusions also provide valuable insights for future adaptive MoE system designs. Our contributions are summarized as follows:

- **An advanced dynamic routing strategy.** We propose a dynamic `Ada-K` routing strategy that adjusts the activated experts number on a per-token basis to enhance the conventional `Top-K` routing. Compared to `Top-K` routing, `Ada-K` routing manages to save over 25% FLOPs and achieve more than 20% acceleration in inference, while enhancing performance.

- **An efficient RL-based training framework.** We introduce a learnable and lightweight allocator module that determines the optimal number of activated experts for each token through end-to-end training. Reinforcement learning techniques have been carefully designed and introduced to facilitate the training of this non-differentiable decision-making framework.

- **Comprehensive quantitative and qualitative experiments.** We extensively evaluate our method across four popular MoE-based LLMs, ranging from 14.3B to 140B parameters. `Ada-K` routing demonstrates consistent advantages over these baselines. The effectiveness of our proposed method is further validated by extensive qualitative analyses.

## 2 RELATED WORK

### 2.1 MIXTURE-OF-EXPERTS

The sparse Mixture-of-Experts (MoE) layer, which includes a predetermined number of experts and a routing network, is initially introduced to enhance the capacity of deep neural networks for NLP tasks in LSTM models (Shazeer et al., 2017a). This architecture is later extended to Transformers (Lepikhin et al., 2020) and adapted for computer vision (Riquelme et al., 2021; Daxberger et al., 2023), gaining popularity due to its robust scaling properties (Du et al., 2022b; Clark et al., 2022). In the MoE framework, extensive research has focused on refining routing algorithms (Hazimeh et al., 2021; Lewis et al., 2021b; Roller et al., 2021; Zhou et al., 2022). Approaches range from random routing (Zuo et al., 2021) and activating all experts via weighted averages (Eigen et al., 2013), to

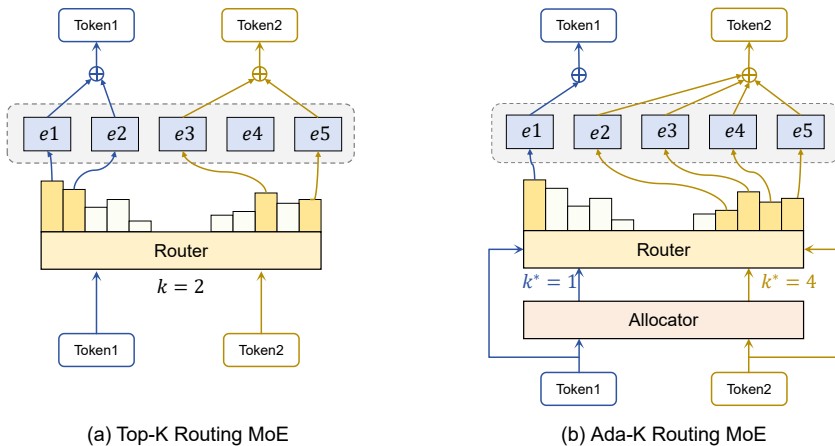

Figure 1: Comparison of `Top-K` and `Ada-K` routing strategies for MoE: (a) In `Top-K` routing, each token consistently selects a fixed number of $k$ experts based on predefined model configuration. (b) In `Ada-K` routing, each token dynamically activates a customized number of experts via our newly introduced learnable allocator module.

selectively engaging a single or multiple experts (Fedus et al., 2022b; Du et al., 2022b). However, they universally deploy a fixed number of experts, irrespective of the differing complexities of input tokens (Lepikhin et al., 2020; Fedus et al., 2022b). In our study, we introduce a parameter-efficient and data-efficient training framework that introduces lightweight learnable modules, *i.e.* allocators. These allocators seamlessly work with routers to dynamically assign expert resources across individual tokens. Our framework markedly reduces computational costs while achieving improved performance compared to the baseline models.

## 2.2 REINFORCEMENT LEARNING IN LLMs

Recent advancements in Large Language Models (LLMs) have been significantly influenced by Reinforcement Learning (RL). For instance, the Reinforcement Learning from Human Feedback (RLHF) approach has demonstrated efficacy in aligning LLMs with human-centric values and preferences (Bai et al., 2022; Ouyang et al., 2022; Cheng et al., 2024). This method involves training a reward model (RM) that encapsulates human preferences and subsequently refining LLMs based on the reward signals generated by the RM. Additionally, some studies (Lee et al., 2023; Xu et al., 2023; Yue et al., 2024) have investigated the LLM-centric multimodal representation learning, utilizing RL to improve the alignment between the semantic spaces of LLMs and visual concepts. Typically, they often leverage the Proximal Policy Optimization (PPO) algorithm (Stiennon et al., 2020; Nakano et al., 2021) to perform parameter optimization. In this work, we present a novel PPO-based training framework tailored for MoE, a widely-used architecture in modern LLMs. This plug-and-play framework seamlessly integrates with MoE models and its variants. We execute end-to-end agents training via RL, with these agents tasked with resource scheduling for expert activation, thereby significantly enhancing the inference efficiency.

## 3 METHOD

### 3.1 PRELIMINARY

We first provide a concise overview of the sparse `Top-K` routing MoE model. Structurally, an MoE layer substitutes the feed-forward network (FFN) sub-block of the original Transformer layer with an expert network comprising $N$ experts $E = \{e_1, e_2, ..., e_N\}$. For each token $x_i$ in the input sequence $X$, the activation probabilities for each expert are determined via a router layer $W$:

$$\mathcal{P}(x_i) = \text{Softmax}\left(W \cdot x_i\right) \tag{1}$$

with $W \in \mathbb{R}^{C \times N}$ being a lightweight, trainable projection matrix. `Top-K` routing MoE employs a routing strategy where the $k$ experts with the highest weights in $\mathcal{P}(x_i)$ are selected. The weights of the chosen experts are then normalized, while that of the remaining experts are set to zero, indicating

their inactivity:

$$g(x_i) = \frac{\mathcal{P}(x_i)}{\sum_{j \in TopK(\mathbf{P})} P_j} \cdot \mathbf{1}_{i \in TopK(\mathbf{P})} \tag{2}$$

where $\mathbf{1}_{i \in TopK(\mathbf{P})}$ is an indicator function that is 1 if $i \in TopK(\mathbf{P})$ and 0 otherwise. The final output, derived from $g(x_i)$, is a weighted average of the $k$ chosen experts:

$$\text{MoE}(x_i) = \sum_{n=1}^{N} g_n(x_i) \cdot e_n(x_i) \tag{3}$$

## 3.2 ADA-K ROUTING

In this section, we introduce the proposed `Ada-K` routing framework outlined in Figure 1. It enhances the conventional `Top-K` routing by incorporating a novel component called allocator. The allocator works synergistically with the original router to dynamically perform expert allocation for each token $x_i$. Structurally, the allocator is a lightweight, trainable linear layer, similar to the router, and is responsible for determining the optimal number of experts $k^*$ for each token. The allocator takes $x_i$ as the input and then produces a probability distribution over the possible number of experts:

$$\mathcal{P}_{alloc}(x_i) = \text{Softmax}(W_{alloc} \cdot x_i) \tag{4}$$

Based on this distribution, $k^*$ is obtained through a non-differentiable sampling operation.

$$k^* \sim \mathcal{P}_{alloc}(x_i) \tag{5}$$

Subsequently, $k^*$ and $x_i$ are passed to the router, which then activates the top $k^*$ experts to produce the weighted representations as described in Sec 3.1. However, direct optimization for these allocators via backpropagation is precluded due to the non-differentiable nature of the sampling operation during the forward pass. To address this challenge, we propose a RL-based optimization framework, which will be detailed in the following section.

## 3.3 LEARNING STRATEGY

In our proposed framework, the entire training is specifically tailored to the newly introduced allocators, with the original LLM maintained in a frozen state to preserve its inherent capabilities.

The training objective is to enhance both performance and efficiency, which can be decomposed into optimizing linguistic capabilities and minimizing the average number of activated experts. Regarding linguistic capabilities, we design a PPO loss, which circumvents the need for differentiable action sampling. Regarding the activated expert counts, we incorporate a regularization loss to minimize the expectation value of the allocator's output distribution.

**PPO Loss.** In the setting of RL, **the allocator of $l$-th layer is considered as an *agent*** with a policy $\pi$ parameterized by $\theta_l$. We introduce a warm-start strategy to initialize $\theta_l$, which will be discussed later. **The representation of $x_i$ at $l$-th layer is regarded as the *state* $s_l$. The number of activated experts $\hat{c}_l$, determined through sampling, serves as the *action* taken under policy $\pi_{\theta_l}$,** *i.e.*, $\hat{c}_l \sim \pi_{\theta_l}(\cdot|s_l)$. The objective is to maximize the expected return $\mathbb{E}[\sum_{l=1}^{L} \gamma^{l-1} R(\hat{c}_l, s_l)]$ over the policy $\pi$, where $\gamma$ serves as the discounted factor, $\gamma \in (0, 1]$, and $L$ represents the number of layers in the model. We define the reward as follows:

$$R(\hat{c}_l, s_l) = \log \mathcal{P}(x_i|x_1, \ldots, x_{i-1}) \cdot \mathbf{1}_{l=L} \tag{6}$$

where only the allocator at the last layer receives the log likelihood as the reward. In other words, the expected return could be simplified as $\mathbb{E}[\gamma^{L-1} \log \mathcal{P}(x_i|x_1, \ldots, x_{i-1})]$. Through the reward defined in this way, maximizing the expected return is equivalent to minimizing caption loss in NLP. We employ the PPO algorithm (Stiennon et al., 2020; Nakano et al., 2021) to optimize the policy within the trust region for stable training. The RL loss function is formulated as follows:

$$\mathcal{L}^{RL}(\theta) = -\mathbb{E}_l \left[\min\left(r(\theta_l)A_l, \text{ clip}\left(r(\theta_l), 1-\epsilon, 1+\epsilon\right) A_l\right)\right] \tag{7}$$

where $\epsilon$ is a hyperparameter and clip function is introduced to constrain values within a specified range. The importance sampling ratio $r(\theta_l)$ is formulated as follows:

$$r(\theta_l) = \pi_{\theta_l}(\hat{c}_l|s_l)/\pi_{\theta_l^{\text{old}}}(\hat{c}_l|s_l) \tag{8}$$

where $\theta_l^{\text{old}}$ is the policy parameters before update. In standard PPO algorithm, it needs another value function to compute the advantage, which requires additional network and computations. Alternatively, we apply the form of advantage function in *reinforce with baseline* algorithm (Sutton & Barto, 2018) formulated in Eq.(9), where the baseline is employed to reduce variance theoretically:

$$A_l(\hat{c}_l, s_l) = \sum_{m=l}^{L} \gamma^{m-l} [R(\hat{c}_m, s_m) - R(c_m^*, s_m^*)] = \gamma^{L-l} [R(\hat{c}_L, s_L) - R(c_L^*, s_L^*)] \tag{9}$$

The superscript * denotes the baseline, which is define as default `Top-K` routing. There is no need of gradients for action $\hat{c}_l$, reward $R(\hat{c}_l, s_l)$, advantage $A(\hat{c}_l, s_l)$, and old sampling probability $\pi_{\theta_l^{\text{old}}}(\hat{c}_l|s_l)$, only latest sampling probability $\pi_{\theta_l}(\hat{c}_l|s_l)$ needs to calculate gradient in training loss.

**Activation Regularization Loss.** The regularization loss reduces the activated expert counts by optimizing the expectation of the distribution produced by every allocator:

$$\mathcal{L}^{reg}(\theta) = \frac{1}{L} \sum_{l=1}^{L} \sum_{n=1}^{N} n \cdot \mathcal{P}_{\theta_l}(n) \tag{10}$$

The final loss is a combination of the PPO loss and regularization loss, where $\lambda$ is a hyper-parameter to control the reduction degree of the activated expert counts.

$$\mathcal{L}(\theta) = \mathcal{L}^{RL}(\theta) + \lambda \mathcal{L}^{reg}(\theta) \tag{11}$$

**Warm Start.** Given the large decision space encompassing all expert counts, allocators require an effective parameter initialization to mitigate instability and inefficiency caused by arbitrary or incorrect choices. In this paper, we propose a warm-start approach "*P-Warm*" to pre-train the allocators. This pre-training process utilizes the nucleus sampling (Holtzman et al., 2019) (*i.e.*, Top-P) to generate pseudo-labels. Specifically, we first choose the minimal subset of experts whose cumulative probability, as determined by the original router, surpasses the threshold $p$. As illustrated in the Eq.(12), for a given token $x_i$, the number of experts within the subset is denoted by $n_i(p)$:

$$n_i(p) = \underset{k \in \{1...,N\}}{argmin} \sum_{j<=k} \mathcal{P}_{i,j}^{\downarrow} \geq p \tag{12}$$

where $\mathcal{P}_{i,j}^{\downarrow}$ represents the probability distribution arranged in descending order. For each baseline, we compute the average expert counts across different $p$ values using a moderate amount of token set $T$. The $p$ value whose average counts is closest to the default activation value is then selected.

$$p^* = \underset{p}{argmin} |\frac{1}{T} \sum_{i=1}^{T} n_i(p) - k|. \tag{13}$$

Finally, for every token $x_j$ in the training dataset, we utilize $n_j(p^*)$ as pseudo labels to facilitate the warm start training of the allocators.

## 4 EXPERIMENTS

### 4.1 IMPLEMENTATION DETAILS

**Model Settings.** We verify the effectiveness and universality of our proposed training framework on four prevailing MoE based LLMs, *i.e.* Mixtral-8x22B (Jiang et al., 2024), Mixtral-8x7B (Jiang et al., 2024), DeepSeek-MoE-16B (Dai et al., 2024) and Qwen1.5-MoE-A2.7B (Team). The architectural details of the four baseline models are presented in Table 1. Mixtral-8x7B and Mixtral-8x22B utilize a standard `Top-K`

Table 1: Architecture details of four baseline models.

| Config | Mixtral 8x22B | Mixtral 8x7B | DeepSeek MoE-16B | Qwen1.5 MoE-A2.7B |
|---|---|---|---|---|
| Top-K | 2 | 2 | 6 | 4 |
| Shared Experts | 0 | 0 | 2 | 4 |
| Routed Experts | 8 | 8 | 64 | 60 |
| MoE Layers | 56 | 32 | 27 | 24 |
| Activated Params | 39.0B | 12.9B | 2.8B | 2.7B |
| Total Params | 140.6B | 46.7B | 16.4B | 14.3B |

routing strategy for all experts. Additionally, DeepSeek-MoE-16B and Qwen1.5-MoE-A2.7B classify experts into shared and routed categories. Each token inherently activates all shared experts and selects the `Top-K` experts from the routed categories. `Ada-K` is applicable to any routing-based expert module. Besides, we keep the shared experts when present.

**Benchmark and Evaluation Details.**    Following previous works (Touvron et al., 2023b; Le Scao et al., 2023; Li et al., 2023; Black et al., 2022), we employ the lm-evaluation-harness (Gao et al., 2021) to evaluate our model. This tool serves as the backend for the HuggingFace Open LLM Leaderboard (Beeching et al., 2023). Our model is assessed on 6 key benchmarks aligned with Open LLM Leaderboard. We firstly examine the model's accuracy across various benchmarks. Then, we evaluate the computational cost with four metrics: average activated expert counts per token (Act), activation reduction rate (Rate), total floating point operations (FLOPs), and inference time speedup (Speedup). "Rate" is calculated as $(1 - Act/k) \times 100\%$, where $k$ represents the default activation value. "Speedup" is the cumulative total across six benchmarks and "FLOPs" is measured by one single sample with a length of 256. Refer to Appendix for additional details.

**Training Details.**    We adopt AdamW (Loshchilov & Hutter, 2017) as the optimizer. All baseline models are trained for one epoch using a consistent set of 10k samples. The batch size and learning rate is set to 64 and 1e-3, respectively. We leverage 2 PPO epochs for reinforcement learning. For all four baseline models, we uniformly set $\lambda$ as 3e-3. This ratio prioritizes ensuring a suf-

Table 2: Training overhead of four baseline models.

| Baseline Model | Total Params | Trainable Params | Training Hours |
|---|---|---|---|
| Mixtral-8x22B | 140.6B | 2.75M | 7.92h |
| Mixtral-8x7B | 46.7B | 1.05M | 4.96h |
| DeepSeek-MoE-16B | 16.4B | 3.54M | 1.79h |
| Qwen1.5-MoE-A2.7B | 14.3B | 2.95M | 1.58h |

ficient reduction rate, while guarantee a performance advantage over the original model. The training overhead for is relatively efficient. As shown in Table 2, the trainable parameters for all baseline models are on the scale of 1M, which is negligible compared to the total parameters. Additionally, we present the training hours for the four baseline models. We employ 16 NVIDIA A800 GPUs to train Mixtral 8x22B, whereas each of the other three utilizes 8 NVIDIA A800 GPUs. The training time for all baseline models is limited to 8 hours. Further details can be found in the Appendix.

## 4.2 PERFORMANCE EVALUATION

Table 3: Performance for four baseline models across six benchmarks. Results using default `Top-K` routing and our method are highlighted in brown and blue, respectively. Each arrow and its associated numeric annotation indicate the performance disparity with the default `Top-K` baseline.

| Method | Accuracy | | | | | | | Computation | | | |
|---|---|---|---|---|---|---|---|---|---|---|---|
| | **ARC-C** | **Hella** | **MMLU** | **GSM** | **Truth** | **Wino** | **Average** | **Act ↓** | **Rate ↑** | **FLOPs ↓** | **Speedup ↑** |
| *Mixtral-8x22B* | | | | | | | | | | | |
| Top-K ($k=1$) | 66.22 | 85.43 | 73.15 | 70.96 | 61.54 | 82.77 | 73.35 ↓5.80 | 1.00 | 50.0% | 11.38T | 1.39× |
| Top-K ($k=2$) | 72.70 | 89.08 | 77.77 | 82.03 | 68.14 | 85.16 | 79.15 | 2.00 | 0.0% | 20.04T | 1.00× |
| Ada-K | 73.57 | 89.76 | 78.22 | 82.98 | 69.97 | 85.02 | 79.92 ↑0.77 | 1.31 | 34.4% | 14.08T | 1.31× |
| *Mixtral-8x7B* | | | | | | | | | | | |
| Top-K ($k=1$) | 60.41 | 83.13 | 64.71 | 40.71 | 34.67 | 75.77 | 59.90 ↓7.68 | 1.00 | 50.0% | 3.68T | 1.35× |
| Top-K ($k=2$) | 66.72 | 86.48 | 70.39 | 58.38 | 41.25 | 82.24 | 67.58 | 2.00 | 0.0% | 6.56T | 1.00× |
| Ada-K | 68.49 | 87.23 | 70.40 | 58.61 | 42.85 | 81.45 | 68.19 ↑0.61 | 1.40 | 30.0% | 4.42T | 1.28× |
| *Qwen1.5-MoE-A2.7B* | | | | | | | | | | | |
| Top-K ($k=2$) | 51.88 | 77.99 | 59.20 | 14.21 | 38.87 | 70.11 | 52.04 ↓2.39 | 2.00 | 50.0 % | 0.88T | 1.25× |
| Top-K ($k=3$) | 52.78 | 78.69 | 60.65 | 14.78 | 40.96 | 72.20 | 53.34 ↓1.09 | 3.00 | 25.0 % | 1.00T | 1.14× |
| Top-K ($k=4$) | 54.69 | 79.45 | 61.30 | 15.62 | 42.50 | 73.00 | 54.43 | 4.00 | 0.0% | 1.23T | 1.00× |
| Ada-K | 54.41 | 79.65 | 60.99 | 21.21 | 41.96 | 72.53 | 55.13 ↑0.70 | 2.58 | 35.5% | 0.92T | 1.22× |
| *DeepSeek-MoE-16B* | | | | | | | | | | | |
| Top-K ($k=3$) | 51.37 | 78.33 | 40.11 | 12.59 | 30.16 | 72.22 | 47.46 ↓2.45 | 3.00 | 50.0% | 0.96T | 1.27× |
| Top-K ($k=4$) | 52.73 | 79.44 | 43.04 | 14.86 | 30.34 | 72.88 | 48.87 ↓1.04 | 4.00 | 33.3% | 1.08T | 1.18× |
| Top-K ($k=5$) | 52.39 | 79.71 | 44.00 | 15.30 | 30.92 | 73.72 | 49.34 ↓0.57 | 5.00 | 16.7% | 1.20T | 1.08× |
| Top-K ($k=6$) | 52.22 | 79.84 | 44.72 | 16.45 | 31.07 | 75.14 | 49.91 | 6.00 | 0.0% | 1.36T | 1.00× |
| Ada-K | 53.44 | 80.24 | 44.98 | 16.86 | 31.83 | 75.92 | 50.55 ↑0.64 | 3.61 | 40.0% | 1.00T | 1.24× |

For the four baselines, the relevant accuracy and computation metrics (detailed in Sec 4.1) are reported in Table 3. Additionally, we present the metrics of the baseline models under a lower `Top-K` activation level (*e.g.*, $k = 1$ for Mixtral-8x7B) as a supplementary comparison. The implementation of `Ada-K` routing leads to a significant and consistent improvement in accuracy across all baselines compared to the default settings. This enhancement is particularly noteworthy, as models employing `Ada-K` routing not only achieve these accuracy gains but also reduce FLOPs by over 25% and accelerate inference by more than 20%, demonstrating a compelling balance between performance and efficiency. We believe these results arises from a more effective expert resource allocation.

### 4.3 ABLATION STUDY

In this section, we perform a comprehensive ablation analysis of the proposed training framework. It should be noted that the conclusions are consistent across four baseline models. However, due to space limitations, we uniformly present the numerical results based on Qwen1.5-MoE-A2.7B.

***Trade-off between performance and activation reduction rate.*** We first investigate the trade-off between the activation reduction rate and model performance. As detailed in Sec 4.1, the results in Table 3 are based on a fixed value of $\lambda$ (*i.e.*, 3e-3) in Equation 11. Actually, Ada-K could facilitates a more flexible balance between accuracy and computational efficiency through it. By varying $\lambda$, we generate 15 data points to construct the trade-off curve illustrated in Figure 2. The trade-off point in the figure correspond to Table 3. Analysis reveals that, prior to the trade-off point, accuracy declines gradually with increasing activation reduction. However, beyond this point, the rate of decline accelerates sharply. Importantly, our method consistently surpasses the default Top-K model until the activation

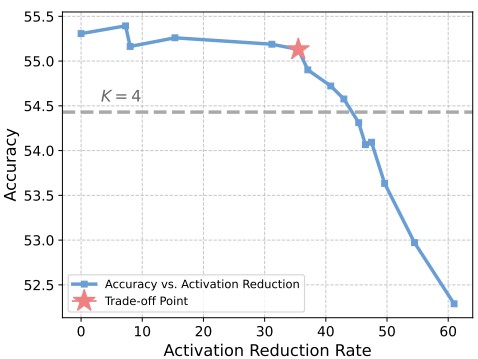

Figure 2: Trade-off curve between performance and activation reduction rate.

reduction rate reaches approximately 44%. Compared to traditional Top-K routing, Ada-K provides both adjustable flexibility and a more effective balance between performance and efficiency.

***Effect of dynamic routing.*** To the best of our knowledge, Ada-K is the first learnable dynamic expert allocation strategy. In this section, we aim to explore and validate the advantages of this pure dynamic paradigm. Actually, as shown in Table 3, at comparable activation reduction rates, baseline models with lower Top-K activation levels exhibit a significant performance gap when compared to Ada-K routing. To adapt these baseline models, which have reduced numbers of active experts, to their new activation patterns, we fine-tune their routers using the same 10K dataset as Ada-K. This process effectively imparts a pseudo-dynamic quality to the static Top-K routing. Additionally, some prior works (*e.g.*, MoE-D (Huang et al., 2024) and D2DMoE (Piórczyński et al., 2023)) achieve quasi-dynamic routing through fixed thresholds. They determine the activation of each expert by comparing the routers' output against these thresholds. We

Table 4: Ablation study about the dynamic routing. The results of default Top-K routing and our method are highlighted in brown and blue, respectively.

| Route | Tuned | Acc | Rate |
|---|---|---|---|
| Top-K ($k=2$) | ✗ | 52.54 ↓1.89 | 50.0% |
| | ✓ | 52.85 ↓1.58 | 50.0% |
| Top-K ($k=3$) | ✗ | 53.34 ↓1.09 | 25.0% |
| | ✓ | 53.44 ↓0.99 | 25.0% |
| Top-K ($k=4$) | ✗ | 54.43 | 0.0% |
| | ✓ | 53.89 ↓0.54 | 0.0% |
| MoED ($p=0.3$) | ✓ | 53.45 ↓0.98 | 32.4% |
| MoED ($p=0.4$) | ✓ | 53.60 ↓0.83 | 28.6% |
| D2D ($\tau=0.1$) | ✓ | 53.73 ↓0.70 | 27.8% |
| D2D ($\tau=0.2$) | ✓ | 53.64 ↓0.79 | 31.5% |
| Ada-K | ✓ | 55.13 ↑0.70 | 35.5% |

also include these methods in our comparison by reproducing them using their official code. The results in Table 4 suggest that improving the adaptability of static routing leads to modest performance gains relative to the original baselines. Additionally, threshold-based routing demonstrates a slight advantage over static routing. However, these methods still fall significantly short compared to Ada-K, highlighting the critical need for a dynamic and efficient expert allocation strategy.

***Effect of activation regularization.*** In this section, we explore different schemes for reducing expert activations. The related results are reported in Table 5. Since the expectation of each allocator's output distribution is differentiable, it allows for direct optimization via backpropagation (row "As Loss"). So we choose it as the default method. Alternatively, combining the language modeling likelihood and the activate expert count of each token to form a reward presents another viable strategy (row "As Reward"). It introduces a reward-level trade-off

Table 5: Ablation study about the activation regularization methods. The results of default Top-K routing and our method are highlighted in brown and blue, respectively.

| Method | Acc | Act | Rate |
|---|---|---|---|
| *Original* | 54.43 | 4.00 | 0.0% |
| As Reward | 54.64 ↑0.21 | 2.56 | 36.0% |
| As Loss | 55.13 ↑0.70 | 2.58 | 35.5% |

where all training objectives are uniformly optimized through the PPO loss. For fair comparison, we employ the same training data and ensure consistent activation reduction rates. The results indicate that direct optimization of expectations yields marginally better performance than treating them as rewards. However, the difference is not significant. It highlights the effectiveness and robustness of `Ada-K` routing training.

***Effect of training data.*** In this section, we examine the impacts of varying training data, *i.e.* pretrain and supervised fine-tuning (SFT) data. For fairness, each data type comprises 10k samples sourced from prominent open-source corpus. We report the training results in Table 6. The results indicate that the training is not sensitive to the data domain. It achieves comparable performance across two data settings, demonstrating both the effectiveness and robustness of `Ada-K` routing training. We detail the collection and organization processes for two types of data in the Appendix.

Table 6: Ablation study about the training data type. The results of default `Top-K` routing and our method are highlighted in brown and blue, respectively.

| Data Type | Acc | Act | Rate |
|---|---|---|---|
| *Original* | 54.43 | 4.00 | 0.0% |
| Pretrain | 55.78 ↑1.35 | 2.68 | 33.0% |
| SFT | 55.13 ↑0.70 | 2.58 | 35.5% |

***Effect of warm up strategy.*** In this section, we conduct an ablation study on different warm start strategies. The results are reported in Table 7. If no warm-up strategy is employed, the allocators are initialized randomly (the second row). The *K-Warm* strategy involves pretraining the allocators to consistently output $k$. For *P-Warm*, as detailed in Sec 3.3, pseudo-labels are generated using a specific $p$ value. The findings indicate that both the *K-Warm* and *P-Warm* strategies obviously surpass the performance of the default baseline, whereas the random strategy shows only comparable performance. It is largely due to the pre-training mitigates the sampling arbitrariness caused by random initialization, facilitating the learning of more optimal strategies. Additionally, the more flexible warmup strategy *P-Warm* yields marginally improved performances compared to the static *K-Warm*.

Table 7: Ablation study about the warm up strategy. The results of default `Top-K` routing and our method are highlighted in brown and blue.

| Strategy | Acc | Act | Rate |
|---|---|---|---|
| *Original* | 54.43 | 4.00 | 0.0% |
| ✗ | 54.18 ↓0.25 | 2.88 | 28.0% |
| *K-Warm* | 54.97 ↑0.54 | 2.60 | 35.0% |
| *P-Warm* | 55.13 ↑0.70 | 2.58 | 35.5% |

## 4.4 VISUALIZATION AND ANALYSIS

In this section, we provide a detailed analysis and visualization of the policies learned by the agents (*i.e.*, allocators) after training with `Ada-K` routing. Due to space limitations, we uniformly present the numerical results based on Qwen1.5-MoE-A2.7B.

***Expert resource allocation is more adaptive.*** We initially analyze the probability distribution of activated expert counts per token: $\mathcal{P}(k) = \frac{r(k)}{\sum_{n=1}^{N} r(n)}$, where $r(k)$ represents the number of tokens activating $k$ experts. The results for each benchmark are illustrated in Figure 3 using a logarithmic scale for enhanced clarity. The analysis reveals: (1) The decision space available to trained allocators ranges from 1 to the total number of experts, ensuring adaptive resource allocation. (2) The training effectively reduces the activation levels of the majority of tokens to approximately 2 or 3. Concurrently, around 10% of essential tokens are identified and subjected to enhanced processing by engaging more than the default number of experts. (3) The expert activation distribution differs across benchmarks, demonstrating that the trained allocators are capable of adapting to diverse domains and devise customized solutions accordingly.

***Middle layers tend to activate more experts.*** As illustrated in Figure 4, variations in layer depth consistently impact average expert activations across multiple benchmarks. Specifically, both the shallow and deep layers utilize fewer experts, whereas the intermediate layers employ a higher number of experts. We hypothesize that this can be attributed to the varying complexities at different stages of processing. Shallow layers mainly engage in basic feature extraction, *e.g.* recognizing simple syntactic patterns and semantic elements, which might generally necessitates minimal specialized knowledge and thus reduces the requirement for experts. Conversely, middle layers might play a pivotal role in more intricate tasks, including the integration of basic features into sophisticated

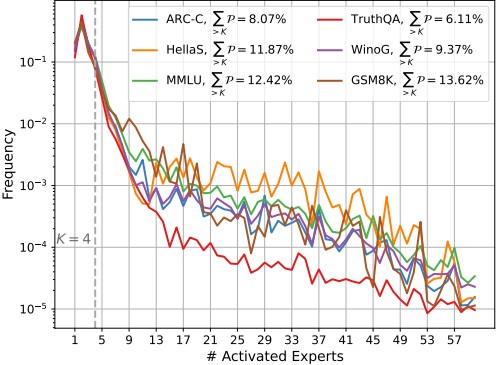 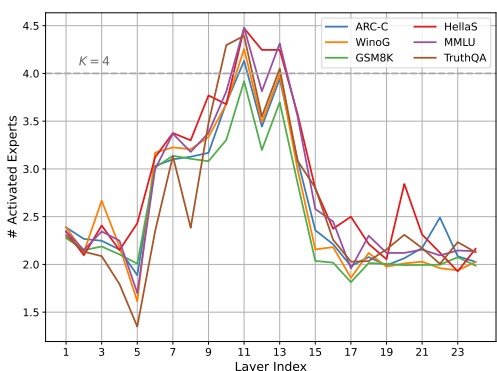

Figure 3: Probability distribution curves of expert activations per token across six benchmarks.

Figure 4: Curves of expert activations per layer across six benchmarks.

representations, ambiguity resolution, and the comprehension of contextual nuances. These tasks could likely benefit from a diverse array of specialized inputs, possibly necessitating a larger pool of experts to enhance the processing robustness. In contrast, deep layers might primarily concentrate on refining these integrated features and finalizing the output, mainly involving the application and optimizing of processed information, which can typically be achieved with fewer experts.

***Content words tend to activate more experts.*** To evaluate the impact of token attributes, we statistically analyze both part-of-speech (POS) and expert activations for a total of 51 million tokens across six benchmarks. To reduce variability due to randomness, our analysis only includes tokens that occur more than 1,000 times within these corpora. POS tagging is performed using the NLTK toolkit (Bird, 2006). We calculate average activations for prevalent POS categories, namely nouns, verbs, conjunctions, adjectives/adverbs, and punctuation. The findings presented in Figure 5 suggest that enhanced expert resource tends to concentrate on verbs and nouns, which are central to syntactic construction and convey clear semantic meanings. In contrast, the expert activations on elements with weaker semantic content, *e.g.* special symbols and conjunctions, are relatively less. This conclusion largely substantiates the intuitive basis for our research motivation. The strategies learned by the agents (*i.e.*, allocators) tend to allocate more expert resources to tokens rich in semantic information, allowing for thorough modeling. Conversely, tokens with weaker semantics require only minimal expert resources for effective representation. This more rational resource allocation strategy benefits both semantic understanding and computational efficiency.

***Load balance is maintained.*** In this section, we investigate whether the previously established expert load balance of the default `Top-K` model is maintained after fine-tuning with our `Ada-K` routing. Intuitively, since the router that determines expert allocation remains frozen throughout the training process, our framework is anticipated to minimally disturb the original load balance. We assess the activation probability of each expert across six benchmarks, as illustrated in Figure 6. The results demonstrate that the load on each expert remains nearly unchanged before and after training, preserving a nearly uniform distribution.

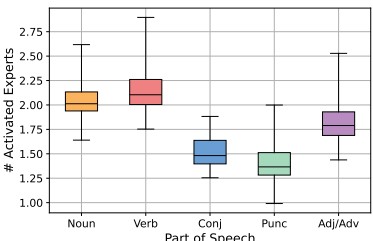 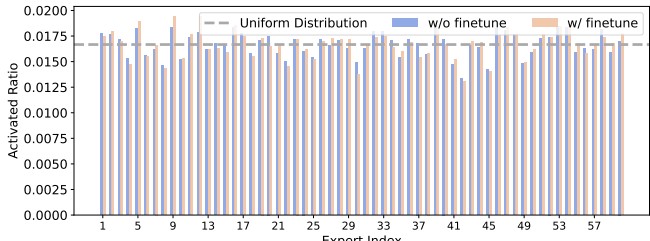

Figure 5: Average expert activation number per token across different parts of speech.

Figure 6: Distribution curves of expert workloads before and after `Ada-K` fine-tuning. The gray horizontal line represents the uniform distribution of expert activations.

Table 8: Average activated experts and accuracy delta across tasks of varying difficulty levels. ΔAcc represents the absolute performance improvement compared to default `Top-K` routing.

| Benchmarks | Act | ΔAcc |
|---|---|---|
| ARC-E | 2.23 | -0.40 |
| ARC-C | 2.84 | +0.12 |
| Collection | 2.58 | +0.70 |
| BBH | 3.43 | +5.54 |

Table 9: The relationship between the ratio of layers equipped with the allocator to all layers and the corresponding metrics. The results with default setting are highlighted in blue.

| Ratio | Training Params | Training Time | Acc | FLOPs |
|---|---|---|---|---|
| 0.125 | 0.37M | 1.54h | 54.98 | 1.19T |
| 0.25 | 0.74M | 1.55h | 55.06 | 1.14T |
| 0.5 | 1.48M | 1.56h | 55.15 | 1.04T |
| 1.00 | 2.95M | 1.58h | 55.13 | 0.92T |

***Hard tasks tend to activate more experts.*** As shown in Table 8, we examine the effect of task difficulty on activated expert counts across both intra-benchmark and inter-benchmark dimensions. For both dimensions, we report the average number of activated experts and the performance gains (relative to the default `Top-K` setting) of our method. For intra-benchmark dimension, we leverage ARC, a multiple-choice question-answering dataset consisting of science exam questions for grades 3 to 9, we analyze two levels of task difficulty, namely Easy (E) and Challenge (C). For inter-benchmark dimension, we compare the commonly used six benchmarks (detailed in Sec 4.1) with BBH (BIG-Bench Hard) (Suzgun et al., 2022). BBH comprises 23 demanding tasks that require sophisticated cognitive skills such as multi-hop reasoning, causal inference, and logical deduction, markedly exceeding the difficulty of the collection of six benchmarks. The results indicate that model with `Ada-K` routing activates more experts on harder tasks (*i.e.*, 2.23 *vs.* 2.84 for intra-benchmark and 2.58 *vs.* 3.43 for inter-benchmark).

Furthermore, we discover that the model with `Ada-K` excels at handling challenging tasks. Specifically, for the intra-benchmark dimension, our method demonstrates a moderate performance loss compared to the baseline with `Top-K` routing in ARC-E. However, when the task difficulty is increased to ARC-C, the model with `Ada-K` exhibits a performance advantage. Similar conclusions are even more evident in the inter-benchmark dimension. This indicates that, when faced with highly complex tasks, the flexibility of `Ada-K` in concentrating expert resources on key tokens enables more effective modeling of the tasks, leading to improved adaptability.

***Equipped more layers with allocator yields better performance.*** In this section, we investigate the impact of varying allocator deployment ratios. By default, we integrate an allocator at each layer of the baseline models. The relevant results are presented in Table 9. All selected layers are sampled at equal intervals across the model. Our findings indicate that, due to the lightweight nature of the allocators, even when all layers are equipped with them, the number of trainable parameters remains minimal, resulting in only a negligible increase in time overhead. Moreover, while this effect does not significantly impact accuracy in benchmark evaluations, equipping more layers with an allocator clearly reduces computational overhead. Consequently, deploying allocators on a per-layer basis emerges as the optimal choice, as it enables a more refined expert allocation strategy at each layer, facilitating the scheduling of expert computations throughout the model.

## 5 CONCLUSION

In this paper, we present a novel `Ada-K` routing strategy for MoE-based LLMs, which dynamically adjusts the number of activated experts based on token importance. `Ada-K` routing enhances the balance between computational efficiency and model performance through a lightweight allocator module optimized via PPO algorithm. Extensive evaluations demonstrate that `Ada-K` routing significantly reduces expert activation by 30%-40% while improving benchmark performance across various MoE-based LLMs. This reduction in expert activation directly translates to computational gains, achieving more than a 25 % reduction in FLOPs and a 20% speedup in inference time, showcasing practical efficiency gains for large-scale applications. Moreover, this method is highly efficient, requiring minimal additional training, and can be easily integrated into existing MoE-based LLMs with low training overhead. Our analysis also provides valuable insights into adaptive resource allocation in large MoE models.

## 6   ACKNOWLEDGMENT

This research is supported by Artificial Intelligence National Science and Technology Major Project (2023ZD0121200), National Natural Science Foundation of China (No. 62437001, 62436001), Strategic Priority Research Program of the Chinese Academy of Sciences (XDA0460305), Key Research and Development Program of Jiangsu Province under Grant BE2023016-3, and CCF-Tencent Rhino-Bird Open Research Fund.

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

## A    EVALUATION BENCHMARKS

Following previous works (Team; Jiang et al., 2024; Dai et al., 2024), we assess models across six standard benchmarks utilizing the Eleuther AI Language Model Evaluation Harness, a comprehensive framework designed to evaluate generative language models on a diverse array of tasks. These tasks include AI2 Reasoning Challenge (ARC-C) (Clark et al., 2018), HellaSwag (Hella) (Zellers et al., 2019), MMLU (Hendrycks et al., 2020), TruthfulQA (Truth) (Lin et al., 2021), Winogrande (Wino) (Sakaguchi et al., 2021) and GSM8K (GSM) (Cobbe et al., 2021). In these evaluations, higher scores indicate better performance. We selected these benchmarks because they assess a range of reasoning abilities and general knowledge across multiple disciplines in both zero-shot and few-shot scenarios.

Table 10: Details of benchmarks. We follow the setting of HuggingFace Open LLM Leaderboard.

| Benchmark | #shots | # Samples | Details |
|---|---|---|---|
| ARC-C (Clark et al., 2018) | 25 | 2.59k | A set of grade-school science questions. |
| HellaS (Zellers et al., 2019) | 10 | 70k | A test of commonsense inference, which is easy for humans but challenging for SOTA models. |
| MMLU (Hendrycks et al., 2020) | 5 | 14.9k | A test to measure a text model's multitask accuracy. The test covers 57 tasks including elementary mathematics, US history, computer science, law, and more. |
| GSM8K (Cobbe et al., 2021) | 5 | 8.5k | Diverse grade school math word problems to measure a model's ability to solve multi-step mathematical reasoning problems. |
| TruthQA (Lin et al., 2021) | 0 | 0.8k | A test to measure a model's propensity to reproduce falsehoods commonly found online. |
| WinoG (Sakaguchi et al., 2021) | 5 | 44k | An adversarial and difficult Winograd benchmark at scale, for commonsense reasoning. |

## B    IMPLEMENTATION DETAILS

In this section, we detail the training protocols of the proposed framework. The specific hyper-parameter configurations for training are reported in Table 11. These protocols are applicable to all the baseline models, utilizing 8 A800-80G GPUs. Throughout all training phases, we consistently conduct a single epoch to prevent overfitting. The batch size per GPU is set at 8. Gradient checkpointing is activated for both training phases to enhance memory efficiency.

Table 11: Additional training details.

| Configuration | Fine-tuning | |
|---|---|---|
| | Warm-Start | PPO |
| Optimizer | AdamW | AdamW |
| Base LR | 1e-3 | 1e-3 |
| Precision | bf16 | bf16 |
| Weight Decay | 0.1 | 0.1 |
| Batch Size | 64 | 64 |
| LR Decay Schedule | cosine | constant |
| Gradient Checkpoint | True | True |
| Training Epochs | 1 | 1 |
| Max Length | 2048 | 2048 |
| Threshold $p$ | 0.3 | – |
| Regularization Coef | – | 3e-3 |
| PPO Epoch | – | 2 |

## C    CHOICE OF $\lambda$

In this section, we present the trade-off curves obtained by scanning $\lambda$ for the remaining three baseline models. The patterns observed are similar: as the compression rate increases, the accuracy initially decreases gradually, but the rate of decline accelerates thereafter. We uniformly select $\lambda = 3 \times 10^{-3}$ as the optimal balance point across all models.

## D    TRAINING CURVES

In this section, we present the advantage and loss curves of four baseline models during training, which are calculated according to Eq.(9) and Eq.( 11), respectively. Generally, a consistent trend

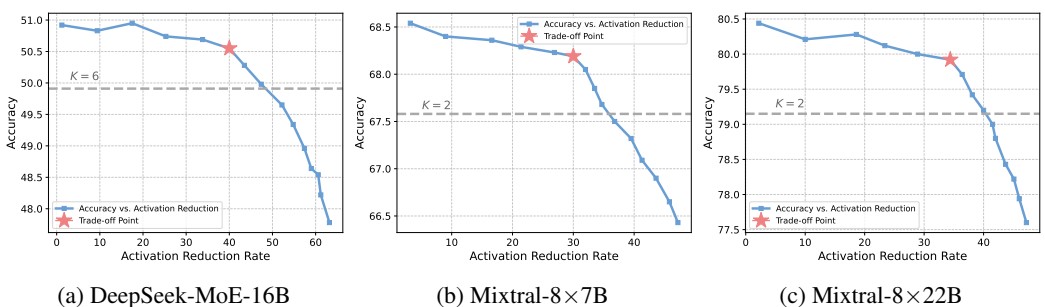

(a) DeepSeek-MoE-16B     (b) Mixtral-8×7B     (c) Mixtral-8×22B

Figure 7: Trade-off curve between perfor- mance and activation reduction rate for other three baseline models.

across all four MoE models is observed: a gradual decrease in loss and an increase in advantage over the course of training. This suggests that the models effectively explored and adopted more optimal strategies, leading to higher rewards.

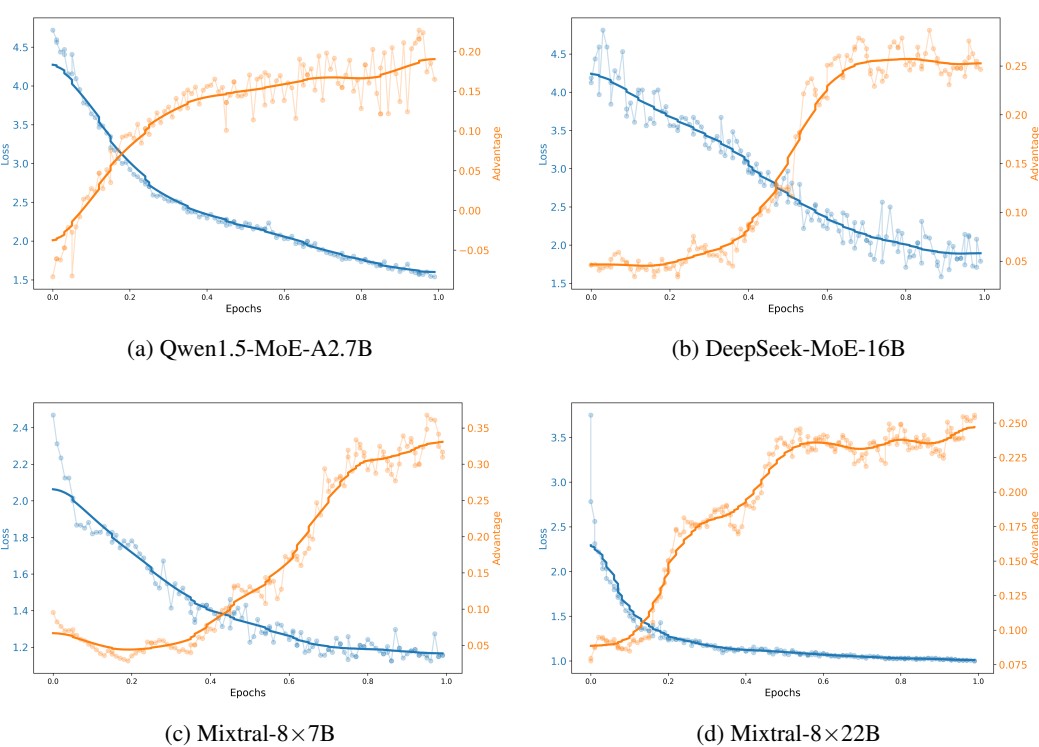

(a) Qwen1.5-MoE-A2.7B     (b) DeepSeek-MoE-16B

(c) Mixtral-8×7B     (d) Mixtral-8×22B

Figure 8: Loss and advantage curves for four baseline models

## E    TRAINING DATA

Table 12: Datasets used for training. We collect data from various sources to empower the model with a broad spectrum of linguistic capabilities. We ensure that all datasets are publicly available in the community.

| Usage | Source | #Sample |
|---|---|---|
| Supervised Fine-tuning | Alpaca GPT4, UltraChat 200k, LIMA, OpenPlatypus CodeAlpaca 20k, Wiki QA, MathInstruct | 10.2k |
| Pretrain | Wiki Demo, RedPajama V2, Wikipedia, StarCoder | 10.1k |

In this section, we introduce the data utilized in our training. For default setting, we employ a dataset comprising 10k randomly sampled from mainstream public supervised fine-tuning datasets. Additionally, the sources of the pretrain corpora used in the ablation experiments are also reported in Table 12. All these data are widely employed in the training of prevalent LLMs (Touvron et al., 2023a; Taori et al., 2023; Chiang et al., 2023).

