# OpenReview forum: "Ada-K Routing: Boosting the Efficiency of MoE-based LLMs"
_ICLR.cc/2025/Conference — ICLR 2025 Poster_

### Official Review · Reviewer_6jjx · 2024-10-29

**Soundness:** 4
**Presentation:** 3
**Contribution:** 4
**Rating:** 8
**Confidence:** 4

**Summary:**

This paper presents Ada-K routing, an additional adapter on mixture-of-expert model that can decide the number of experts activated for each token. The dynamic number of experts helps reducing the inference cost of the model, while still keeping a descent performance. The authors used proximal policy optimization algorithm to make the ada-K adapter trainable, and evaluated it on four MoE models at different scales. The authors additionally run a sweep of experiments to study the effect of Ada-K at different scale of MoE, the trade-off between accuracy and efficiency, and the ablation study of different hyper-parameters. The authors further introduced visualization of the expert allocation pattern, which helps understand the MoE architecture's dynamism.

**Strengths:**

- The idea of an adapter to control the number of experts at the granularity of token and layer can be universally migrated to most of the MoE models, and is simple to implement.
- The paper presents detailed optimizations to make the reinforcement learning of adapter more stable.
- The experiment is abundant and at a large scale, including:
  - models at different scale of both parameters and activated parameters, as well as different baseline number of activated experts.
  - the pareto frontier between performance and the cost reduction
  - comparing Ada-K with existing heuristic-based dynamic expert allocation algorithms
  - ablation study on dataset, regularization, and warmup strategy
- The visualization of the result is novel and inspiring.

**Weaknesses:**

- Although Ada-K significantly reduces the total FLOPs at inference, an efficient implementation could be nontrivial, which limits the use of Ada-K technique in practice. Providing experiment data on the average latency/throughput could help make the inference cost improvement more significant.
- Figure 2 suggests that the hyperparameter $\lambda$ is crucial to the balance between performance maintenance and the inference cost reduction. However, the corresponding paragraph ("Trade-off between performance and activation reduction rate") lacks enough information on how to find the $\lambda$. What is the range of $\lambda$ in the figure? Is the best $\lambda$ generally applied to all the four models, or should a user of Ada-K also finetune on a sweep of different $\lambda$'s? Providing a plan for the user to find a good $\lambda$ could help improve Ada-K's usability.

**Questions:**

- I noticed that the authors claim that all conclusions in the ablation study is generally applied to all the four models. Providing the trade-off point of all other three models (or are they also 3e-3?) could answer my doubt on the second point of the weakness section.

The paper is well written in general. The question below will not influence my scoring.
- In section 3.3, paragraph PPO Loss, "Eq.(4) could be simplified as ...". I'd suppose it is "The expected return could be simplified as ..."

---

> ### Author Response · Authors · 2024-11-20
> **Rebuttal by Authors**
>
> Dear Reviewer 6jjx,
>
> We sincerely appreciate your valuable and insightful comments. We found them extremely helpful for improving our manuscript. We will strive to address each comment in detail, one by one below.
>
> ---
>
> **W1. Latency & Throughput**
>
> Thank you for your insightful suggestion. In response, we wish to address your concern in the following two aspects:
>
> * Following your valuable guidance, we conduct additional tests on throughput and latency under inference settings, specifically using a NVIDIA A800 GPU with a total batch size of 16 and max new length of 64 based on Qwen1.5-MoE-A2.7B. To minimize randomness, we randomly select 16 prompts from a pool and repeat the test for five times.
>     * For throughput：we measure the number of tokens processed per second (including both input and output tokens).
>     * For latency： We measure the time (second) from inputting a prompt to receiving a response.
>
>   Due to time constraints, we will include the complete results for all four baseline models in Table 3 in the next version of the paper.
>
> * In addition to FLOPs, we kindly request your attention to Speedup, which represents the reduction in total inference time across all benchmarks after implementing Ada-K. This metric reflects the advantages of Ada-K when deployed in practical applications.
>
> We summarize all the metrics in the table below to provide a clear visualization of the inference acceleration effects achieved by Ada-K.
>
>   |   Method | Avg Acc | Act↓ | Rate↑ | FLOPs↓ | Speedup↑ |  Throughput↑ | Latency↓ |
>   | :---------- | :-----: | :-----: |:-----: |:-----: |:----: |:----: |:-----: |
>   | Top-K (k = 2) | 52.04 |2.00 | 50.0 % |  0.88T  |  1.25× | 51.06 | 9.35 |
>   | Top-K (k = 3) | 53.34 |3.00 | 25.0 % | 1.00T |  1.14× | 48.63 | 9.81 |
>   | Top-K (k = 4) | 54.43 |4.00 | 0.0% | 1.23T | 1.00× | 41.48 | 11.38 |
>   | Ada-K |55.13| 2.58 | 35.5%  | 0.92T |1.22× | 50.17 |  9.52 |
>
>
> **W2 & Q1. Choice of $\lambda$**
>
> * **Balanced Point**: Thank you for your thorough consideration. As we introduced in the original manuscript (L286-288), **we set $\lambda = 3e-3$ for all four baseline models**. We empirically find that this value results in similar reduction rates and performance enhancements across all baselines. Therefore, it is adopted as the default setting.
>
>  * **More Results**: The scanning points for $\lambda$ are ranging from 1e-6 to 1. We have reported the trade-off curves for other three baseline models in **Appendix E: CHOICE OF $\lambda$**. We respectfully suggest that you refer to these results for more detailed information.
>
>
> **Q2. Expression Correction**
>
> Thank you for your meticulous review. We have addressed this point in the revised version.

---

> > ### Comment · Reviewer_6jjx · 2024-11-22
> > **Feedback of Rebuttal**
> >
> > Thank you for your response. It addressed all my concerns. I will keep the score.

---

> > > ### Author Response · Authors · 2024-11-23
> > > **Official Comment by Authors**
> > >
> > > Dear Reviewer 6jjx,
> > >
> > > Thank you for your recognition of our work and the time and effort you have invested as a reviewer!
> > >
> > > We will adhere to your valuable suggestions to refine our manuscript accordingly.

---

### Official Review · Reviewer_wxUn · 2024-10-29

**Soundness:** 3
**Presentation:** 2
**Contribution:** 3
**Rating:** 6
**Confidence:** 3

**Summary:**

This paper introduces a new Ada-K routing strategy for MoE-based large language models (LLMs), which dynamically adjusts the number of activated experts based on token importance. Due to the non-differentiable nature of the decision, the paper leverages the Proximal Policy Optimization (PPO) algorithm to facilitate the end-to-end learning process. The proposed method improves model performance while reducing computational costs. Extensive evaluations on various benchmarks, along with comprehensive ablation studies, demonstrate the effectiveness of Ada-K routing compared to traditional Top-K routing strategies.

**Strengths:**

1. The paper investigated a timely problem.
2. The routing strategy proposed in this paper is a new approach that dynamically adjusts the number of activated experts.
3. The paper presents a comprehensive set of experiments and analyses.

**Weaknesses:**

1. The structure of the Method section can be improved. Section 3.1 describes the traditional routing method of MoE, which would be more appropriately placed in the Related Work section. The Method section needs more elaboration. At present, it is rather brief, and expanding on key details would greatly improve the clarity and comprehensibility of the research.
2. This method has certain limitations, particularly in its application to models that use top-1 routing, such as Switch Transformer, making optimization more challenging.
3. Regarding the experiments in Section 4.2, a direct comparison between the warm-up Ada-k routing and the baseline top-k routing with different k values may be somewhat unfair. The models are likely to have different performance levels even before training due to the differences in routing methods. Providing a loss curve during training could better demonstrate the effectiveness of the proposed method.
4. Figure 8 shows that the use of the Ada-k strategy leads to performance degradation on simpler tasks while achieving better results on more complex tasks. This seems counterintuitive based on prior experience. Perhaps the authors could provide a more plausible explanation for this phenomenon.

**Questions:**

In addition to the issues mentioned in the Weaknesses section, there are a few other concerns:

1. What does $i$ represent in Equation (8)? Perhaps it should be $n$?
2. The authors should provide more details on how the policy $\pi$ was designed in this work and the rationale behind this design choice? Additionally, how was the number of training parameters calculated?

---

> ### Author Response · Authors · 2024-11-20
> **Rebuttal by Authors [1/2]**
>
> Dear Reviewer wxUn,
>
> We sincerely appreciate your valuable and insightful comments. We found them extremely helpful for improving our manuscript. We will strive to address each comment in detail, one by one below.
>
> ---
>
> **W1. Method Structure**
>
> Thank you very much for your valuable and constructive comments. Based on your suggestions, we have restructured the sections of our paper, and expanded the Method section, especially emphasizing and detailing key technical aspects. We warmly welcome and greatly appreciate any further suggestions.
>
>
> **W2. Top-1 Routing**
>
> We appreciate the reviewer's valuable insights and would like to clarify the following two points:
> ​
> * **Top-1 compatibility**: We would like to clarify that Ada-K is easily adaptable for scenarios where k=1. Each allocator in our framework currently has a decision space that ranges from activating one expert to activating all experts for a given token.
> To address Top-1 routing, we can simply extend this decision space to include the option of **"selecting 0 experts"**. We conduct related experiments based on Switch Transformer base,  using the same benchmarks originally employed in its paper for fairness:
>
>   |      Method      | XSum↑ | ANLI↑  | ARC-E↑ |  ARC-C↑ | Act↓ | Rate↑ | FLOPs↓ | Speedup↑  |
>   | :-- | :-------: | :-----: |:-----: |:-----: |:-----: | :-----: |:-----: |:-----: |
>   | Top-K (K=1) | 19.1 | 51.4 | 63.9 | 36.5 | 1.00 |  0.0% | 106.01G | 1.00× |
>   | Ada-K |  20.2  | 51.8 | 64.7 | 37.8 | 0.72 | 27.6% | 80.62G | 1.21× |
>
> * **Top-1 application**: We respectfully emphasize that recent mainstream MoE-based LLMs have largely moved away from the Top-1 routing strategy [1,2,3,4]. Instead, these models adopt larger values of k to **unlock more flexible and diverse expert combinations**, which significantly enhance performance.
>
>
> [1] Dai, Damai, et al. "Deepseekmoe: Towards ultimate expert specialization in mixture-of-experts language models." arXiv preprint arXiv:2401.06066 (2024).
>
> [2] Yang, An, et al. "Qwen2 technical report." arXiv preprint arXiv:2407.10671 (2024).
>
> [3] Jiang, Albert Q., et al. "Mixtral of experts." arXiv preprint arXiv:2401.04088 (2024).
>
> [4] Xue, Fuzhao, et al. "Openmoe: An early effort on open mixture-of-experts language models." arXiv preprint arXiv:2402.01739 (2024).
>
> **W3. Model Comparison**
>
> Thank you for your valuable comments. We wish to address your concerns based on the following three points:
> ​
> * **Training Curves**: We have included the advantage and loss curves for the four MoE models in the **Appendix D:TRAINING CURVES**, calculated according to Eq.(9) and Eq.(11), respectively. We respectfully suggest that you refer to these results for more detailed information. Overall, a consistent trend observed across all four MoE models is a gradual decrease in loss and an increase in advantage during training, indicating that the models effectively explored and adopted more optimal strategies to achieve higher rewards.
>
> * **Baseline Performance**: We wish to respectfully clarify an unintended misunderstanding: the performance of these three configurations **originates from the same checkpoint** and **it does not exhibit "*different performance levels even before training*" as you mentioned in the review**. Taking the results of Mixtral-8x7B in Table 3 of Sec 4.2 as an example, the performances for Top-K (K=1) and Top-K (K=2)  are **directly derived from the original checkpoint without any training**, only adjustments to the value of K. Subsequently, we froze this original checkpoint and trained the new allocators with approximately 10K data samples to obtain the performance for Ada-K. Additionally, in the next point, we will discuss a more rigorous and fair comparison.
>
> * **Further Evaluation**: Regarding the comparisons in Table 3, we conduct a more rigorous and fair analysis, as detailed in Table 4 of the original manuscript. Although the original checkpoint for Ada-K training was frozen, we did utilize 10k data samples to train the allocators. To ensure a more fair comparison, we fine-tuned the Top-K (K=1 and K=2) baselines using the same data. We documented the performances before and after tuning in Table 4, distinguishing them with "tuned" indicated as either ✓ or ✗. The results demonstrate that while fine-tuning yields benefits, the performances of Top-K baselines after fine-tuning are still inferior to that of Ada-K.

---

> ### Author Response · Authors · 2024-11-20
> **Rebuttal by Authors [2/2]**
>
> **W4. Simple & Hard Tasks**:
>
> We greatly appreciate the reviewer's detailed observations. Our Ada-K strategy is based on the premise that different tasks have varying complexities, necessitating dynamic adjustment in the number of activated experts accordingly. We wish to address your concerns based on the following two points:
>
> * We would like to respectfully clarify that the "*performance degradation on simpler tasks*" you mentioned applies only to the intra-benchmark setting, *i.e.*, ARC-Easy vs. ARC-Challenge. Given the limited number of samples (\~5k) in the ARC dataset, some performance fluctuation may occur. However, in the inter-benchmark setting, where Collection serves as the simpler task, there are significantly more samples (\~140k) compared to ARC-Easy, which mitigate such fluctuations, achieving an accuracy gain of 0.7%.
>
>
> * There is a trade-off between performance and FLOPs. Gains in performance should be considered **in the context of the corresponding computational load**. Although the performance gains on simpler tasks may not be as significant, the model utilizes fewer expert computational resources. In contrast, when faced with more challenging tasks, the allocators tend to over-allocate resources to certain important tokens, enabling better feature modeling and more accurate responses. This difference bewteen complex and simple tasks highlights the adaptive advantage of dynamic routing over static Top-K routing.
>
> **Q1. Equation (8) Notation**:
>
> Thank you for your meticulous review. As you correctly pointed out, $i$ should be replaced with $n$. We have made the necessary correction in the paper.
>
> **Q2. Policy $\pi$ and Trainable Parameters**:
>
> * The policy $\pi$ in reinforcement learning is a fundamental concept that represents the decision-making strategy an agent (*i.e.*, the allocator) uses to determine actions (*i.e.*, the activated expert numbers of a given token) based on the current state (*i.e.*, the representation of a given token). Concretely, it can be understood as the probability distribution outputted from the allocator when the token representation is inputted. This distribution guides the sampling over possible expert activations, reflecting the tailored computational strategy for each token to optimize performance and efficiency dynamically.
>
> * Each allocator is a single linear layer without bias. As only the allocators are trained, the total number of trainable parameters amounts to $C \times N \times L$, where $C$ is the hidden size, $N$ is the total number of experts, and $L$ is the number of layers.

---

> > ### Comment · Reviewer_wxUn · 2024-11-21
> >
> > Thanks for the response, i raised the score to 6.

---

> > > ### Author Response · Authors · 2024-11-21
> > > **Official Comment by Authors**
> > >
> > > Dear Reviewer wxUn,
> > >
> > > Thank you for your recognition of our work and the time and effort you have invested as a reviewer!
> > >
> > > We will adhere to your valuable suggestions to refine our manuscript accordingly.

---

### Official Review · Reviewer_gGQ7 · 2024-10-29

**Soundness:** 2
**Presentation:** 3
**Contribution:** 3
**Rating:** 6
**Confidence:** 5

**Summary:**

This work proposes to introduce an adaptive computation budget for MoE LLMs. Specifically, the proposed method fine-tunes a pretrained MoE LLM to activate the adaptive number of experts with PPO training and a trainable allocator layer. The proposed model achieves comparable performance with baselines but uses less computation.

**Strengths:**

1) Well-motivated. It is well-known that MoE LLMs are very effective and promising, but the efficiency of MoE LLM deployment is limited due to the huge amount of trainable parameters. It is good to improve the efficiency of LLMs.
2) Clear writing and comprehensive ablation studies.

**Weaknesses:**

1) An important baseline is missing -> Mixture of Depth (https://arxiv.org/abs/2404.02258). Due to the layer skip in this paper, the computation cost for each token is adaptive as well,
2) Due to the imbalanced computation cost in different layers, the pipeline parallelism is more difficult and challenging to use, during both training and inference.
3) There are many other ways to introduce adaptive computation budget, e.g. ACT algorithm in universal transformer (https://arxiv.org/abs/1807.03819), PonderNet (https://arxiv.org/abs/2107.05407). Need to discuss and compare. Why do you select PPO to train the allocator? More justification is required. It seems that the model is unnecessarily complex. Why not just ACT or so? Any algorithm or training difficulty? Introducing the PPO in such an early stage will make the LLM training pipeline much more complicated, which may make this approach not that useful, even if it is effective to some extent.

**Questions:**

1) What is the setting of LLM inference speedup? Batch inference or batch size == 1? How does the inference speed trend if we ablate the batch size? And are you using expert parallelism during training and inference?

---

> ### Author Response · Authors · 2024-11-20
> **Rebuttal by Authors [1/4]**
>
> Dear Reviewer gGQ7,
>
> We sincerely appreciate your valuable and insightful comments. We found them extremely helpful for improving our manuscript. We will strive to address each comment in detail, one by one below.
>
> ---
>
> **W1. MoD Comparison**
>
> Thank you for pointing out this baseline. To address this as comprehensive as possible, we conducted experiments comparing Ada-K with three MoD-like variants. Since the official code is not open source, we refer to the two most famous reproductions: https://github.com/astramind-ai/Mixture-of-depths and https://github.com/kyegomez/Mixture-of-Depths.
>
> - **Expert-Level MoD**：Each expert is assigned a binary gate. Each binary gate is used to decide whether each token should bypass the corresponding expert.
> - **MoE-Level MoD**：Each MoE sublayer is assigned a binary gate. Each binary gate is used to decide whether each token should bypass the corresponding MoE sublayer.
> - **Layer-Level MoD**: It is the classic MoD design, where a new gate is introduced to decide whether each token should bypass the corresponding Transformer layer (including both the Self-Attention and MoE sub-layers).
>
> The comparison results based on Mixtral-8x7B are summarized in the table below. For fair comparison, we adopt the same training and data setting, and introduce the MoD gate at each layer. We set the capacity, which is a hyperparameter used in MoD to decide whether to skip computations, for the three variants to ensure they have similar FLOPs to Ada-K. It enables a fair comparison of average accuracy.
>
>    | Method | Avg Acc↑ | Act↓ |FLOPs↓ |
>   |:--|:--:|:--:|:--:|
>   | Expert-Level MoD |  65.47 | 1.43 |4.58T |
>   | MoE-Level MoD |   64.96 | 1.39 |4.41T |
>   | Layer-Level MoD | 62.42 | 1.38 | 4.36T |
>   | Ada-K |  68.19 |  1.40 |4.42T |
>
>   The comparison highlights several key advantages of Ada-K: **(1) Performance Superiority**: Ada-K achieves significantly better performance compared to these MoD variants while maintaining similar FLOPs. **(2) Pure Dynamic Routing Decision**: Unlike MoD-based methods that require pre-defined capacity thresholds, the decision-making process for Ada-K is fully learnable. This eliminates the need for manually setting thresholds for specific scenarios or models, offering significant flexibility and generalizability. **(3) More Reasonable Allocation.**: These MoD gates, when choosing to skip computations, functions similarly to allocating fewer experts to certain tokens in Ada-K. However, the ability of Ada-K to adaptively select critical tokens and allocate them with more expert resources is something that MoD-based methods struggles to offer. This adaptability is a key reason for its superior performance. **(4) Seamless Autoregressive Compatibility**: MoD’s design requires sorting token weights to decide which tokens to skip computations, which is incompatible with autoregressive sampling, as future tokens’ weights are unknown. It introduces the need for additional modules or losses, which come at the cost of performance (as confirmed in the original MoD manuscript).However, Ada-K avoids this issue, making it more suitable for autoregressive LLM.

---

> ### Author Response · Authors · 2024-11-20
> **Rebuttal by Authors [2/4]**
>
> **W2. Pipeline Parallelism Compatibility**
>
> It is indeed a very insightful question. In fact, we also considered similar concerns and proposed a straightforward solution to make our approach compatible with pipeline parallelism. We would like to elaborate on the details in the following two points:
>
> * **Methodology Analysis:** As stated in L220-L226 of our manuscript, we use the regularization loss in Eq.(10), abbreviated as **global loss**, to compress the number of activation experts. This approach provides a global control, as it focuses on the global average mathematic expectation of activation experts. Due to its simplicity, we adopt it as the default strategy. Besides, we have also tried a more granular, layer-specific method, abbreviated as **local loss**, to enhance the model's compatibility with pipeline parallelism:
>
> $$
> \mathcal{L}_l = \frac{1}{|\mathcal{T}_l|} \sum _ {t \in \mathcal{T}_l} \max(0, | \mathbb{E}[p _ {\theta_l}^t] - m | - \delta)
> $$
>
>   For the $l$-th layer, $\mathcal{T} _ l$ denotes the set of tokens. Each token $t$ is processed by the allocator $\theta_l$ which   outputs probability distributions $p _ {\theta_l}^t$ for selecting a certain number of experts. The term $\mathbb{E}[p_{\theta_l}^t]$ is the mathematical expectation of activated experts for token $t$. $m$ specifies the desired number of active experts, and $\delta$ allows a small tolerance around this target. This local loss, akin to a hinge loss, optimizes expert activations by minimizing deviations from $m$. When a uniform $m$ is applied, **it helps balance the computational load across all layers**, enhancing compatibility with pipeline parallelism.
>
> * **Experiment Validation:** We present a comparison between the results of using global loss and the local loss settings in the table below, based on the Mixtral-8x7B with Top-K=1 and Top-K=2 serving as baseline references. Both loss functions are applied under similar FLOPs to ensure fairness. Additionally, We report two variance metrics: Layer Std and Token Std:
>   - Layer Std: This metric measures the variance in the average number of activated experts per layer during inference, reflecting differences in computational load across layers.
>   - Token Std: This metric assesses the variance in the number of experts activated per token across all layers, illustrating the variability in expert allocation for individual tokens.
>
>   |  Method | Trainable Param (M) | Avg Acc (%) | FLOPs (T) | Layer Std | Token Std |
>   | :-- | :-: | :-----: | :---: | :--: | :--: |
>   | Baseline (Top-K = 1) | -- |  59.90  | 3.68 |  0.00  | 0.00 |
>   | Baseline (Top-K = 2) |    --   |  67.58  | 6.56 |   0.00  | 0.00 |
>   | Ada-K + global loss | 1.05 |  68.19  | 4.42 |  0.62  | 0.79 |
>   | Ada-K + local Loss  |  1.05  |  68.08    | 4.39 | 0.07  | 0.75 |
>
> Based on our experimental results, we wish to discuss the following two observations:
>
> * Under the same allocator structure, the local loss slightly underperforms compared to the global loss. However, it still presents a significant efficiency improvement over both Top-K baselines.
>
> * The Layer Std for local loss is substantially lower than that for global loss, indicating that the layer-wise local loss strategy balances computational differences between layers more effectively, facilitating better compatibility for pipeline parallelism. Besides, as shown in Token Std, the seemingly tighter layer-wise constraint has a relatively small impact on token-wise exploration. Each token is able to freely select the number of active experts under both loss functions.

---

> ### Author Response · Authors · 2024-11-20
> **Rebuttal by Authors [3/4]**
>
> **W3. Adaptive Computation Method Comparison**
>
> We appreciate the opportunity to discuss the following three points:
>
> * **Why not just ACT and so?:** ACT and its subsequent work, PonderNet, introduce adaptive computation budgets into single-layer recurrent computations to dynamically adjust computation time steps based on input complexity. They skip unnecessary computations by setting cumulative probabilities or through sampling. Following your suggestion, we have incorporated their idea into token-level adaptive computation. We strictly follow the official implementations of ACT and PonderNet ([ACT](https://github.com/andreamad8/Universal-Transformer-Pytorch), [PonderNet](https://nn.labml.ai/adaptive_computation/ponder_net/index.html)) for experimental comparison. We used the same training and data settings, and hyperparameters are adjusted to compare performance under similar FLOPs constraints.
>
>     |      Method      | Avg Acc (%) ↑ | FLOPs (T) ↓ |
>     | :-------------- | :-------: | :-----: |
>     | Mixtral (Top-K = 2) | 67.58 |	6.56 |
>     | $\quad$ + ACT  |  62.46   | 4.45 |
>     | $\quad$ + PonderNet |  63.75  | 4.38  |
>     |  $\quad$ + Ada-K    |   68.19   |  4.42  |
>
>    Building upon the results, the PPO-based Ada-K demonstrates significant performance advantages. Moreover, we would like to emphasize that in implementing dynamism, both ACT and PonderNet still require **preset thresholds**, making them less flexible than allocators trained with PPO. For a more detailed discussion of the motivations for using PPO, please refer to the next point.
>
>
> * **Why PPO**？
>   Beyond clear performance benefits, the reason we employ PPO-driven allocators is:
>     - For **efficiency**, we aim for the entire training process to be end-to-end to achieve holistic optimal learning. However, since the number of experts assigned to each token is sampled from allocator's output distribution, this sample operation is inherently non-differentiable, making it unrealistic to optimize directly via standard backpropagation.
>     - For **fine-grained**, we incorporate allocators at each layer, enabling it to make both token-specific and layer-specific decisions. The overlay of decisions across layers results in a dynamic and continuous decision-making process, which is highly complex.
>     - For **flexibility**，besides ACT and PonderNet you mentioned, we also compared two other threshold-based adaptive computation methods (*i.e.*, MoED and D2D) in Table 4 of the original manuscript. Ada-K likewise demonstrated the performance advantages. It largely demonstrates that the flexibility shortcomings of this threshold-based dynamic approach, compared to a fully adaptive PPO method, indeed affect performance.
>
>   Considering the above, we employed PPO algorithm, known for **the robustness in complex decision-making**. In this way, the allocators are optimized through **policy gradients**, without the necessity of standard backpropagation.
>
> * **"Unnecessarily complex"?**: We wish to respectfully clarify that our PPO-based Ada-K framework is a concise and efficient design without undue complexity.
>     * **Model Structural Complexity**: Our model simply adds a small linear module to each layer of a fully frozen and pre-trained MoE-based LLM. The total parameter count for these additional modules is only about 1M, which is less than $10^{-4}$ of the LLM's total parameters. Additionally, MoD, ACT, and PonderNet you mentioned also require similar extra modules to determine when to halt computation.
>
>     * **Model Training Complexity**: You mentioned concerns about "*Introducing PPO at an early stage complicating the LLM training pipeline.*" However, we kindly wish to clarify this **misunderstanding**. Actually, Ada-K is a post-training strategy applied to a fully pre-trained and frozen MoE-based LLM, not at an early stage. Moreover, it is not only straightforward but also highly efficient, with all trainings completed within 8 hours (as detailed in Table 2 of our manuscript).

---

> ### Author Response · Authors · 2024-11-20
> **Rebuttal by Authors [4/4]**
>
> **Q1. Inference Speedup**
>
> Thank you for your insightful questions. We would like to clarify the following points in response:
>
> * **Evaluation Setting**: All inference speedup tests are conducted using 8 NVIDIA A800 GPUs, with a consistent total batch size of 16 set for all benchmarks. We **utilize expert parallelism**, with different experts distributed across various GPUs, where each GPU only processed a token group corresponding to the experts on that device.
>
> * **Balanced Expert Load**: Actually, the variance in expert load distribution before and after the Ada-K training is minimal, maintaining a consistent and balanced allocation. This stability is achieved by freezing the original model parameters, particularly the routers responsible for selecting the experts. This visualization is reported in Fig.6 of the original manuscript. In other words, **Ada-K uniformly and fairly reduces the computational load allocated to each expert**. This characteristic is particularly beneficial for batch inference, which we will have a discussion in the following point.
>
> * **Batch Size Ablation**: In response to your guidance, we further conduct an ablation study on inference speed across different batch sizes based on Mixtral-8x7B, detailed in the table below:
>
>    | Batch Size | Speedup   |
>    |------------|-----------|
>    | 1          |  1.248×   |
>    | 4          |  1.267×   |
>    | 16 (default) | 1.284×  |
>    | 32         | 1.288×    |
>    | 64         | 1.285×    |
>
>   The results indicate that at smaller batch sizes, due to fewer tokens per batch, the variability in the number of tokens processed by each expert may be greater, which may introduce randomness into acceleration effect evaluations. However, as the batch size increases, the growth in token counts stabilizes the expert loads towards a uniform distribution. The advantages of Ada-K in uniformly reducing computations for each expert are more consistently demonstrated.

---

> > ### Comment · Reviewer_gGQ7 · 2024-11-21
> >
> > Thanks for the response, i raised the score to 6. Good luck!

---

> > > ### Author Response · Authors · 2024-11-21
> > > **Official Comment by Authors**
> > >
> > > Dear Reviewer gGQ7,
> > >
> > > Thank you for your recognition of our work and the time and effort you have invested as a reviewer!
> > >
> > > We will adhere to your valuable suggestions to refine our manuscript accordingly.

---

### Official Review · Reviewer_3rXb · 2024-11-04

**Soundness:** 3
**Presentation:** 3
**Contribution:** 3
**Rating:** 6
**Confidence:** 4

**Summary:**

This paper introduces the Ada-K Routing method, which incorporates an additional RL agent into pre-trained MoE models to dynamically control the top-k in MoE routing. This approach enables dynamic adjustment of top-k at a lower cost, thereby enhancing the inference efficiency of MoE models. The effectiveness of the proposed method has been validated on multiple open-source models, demonstrating improved accuracy compared to existing threshold-based methods when achieving similar acceleration effects. Additionally, the paper presents detailed ablation studies and analytical experiments that provide valuable insights into the research on MoE models.

**Strengths:**

- Innovatively proposes to introduce an RL agent for controlling top-k in pre-trained MoE models, optimizing the allocation of computational resources and improving inference efficiency. The method has a low training cost and shows robustness to training data, outperforming existing methods across various downstream tasks.
- Thorough ablation studies demonstrate the relationship between acceleration effects and accuracy, providing practical guidance. It also validates the effectiveness of activation regularization and warm-up strategies.
- Performs meticulous analysis revealing that intermediate layers of trained models require the activation of more experts, reaffirming that more challenging tasks necessitate the activation of more experts.

**Weaknesses:**

- Some comparisons with baseline methods are not entirely reasonable. Existing work suggests that freezing the router yields better results when tuning MoE models. Ada-K freezes the router and introduces additional agent parameters for training, while other comparison methods only train the router. The reviewer recommends supplementing the results by (1) conducting full fine-tuning of the model with a frozen router and comparing the effects of introducing only threshold methods versus Ada-K; or (2) maintaining the current Ada-K setup and adding LoRA parameters to the threshold-based method while freezing the router for tuning.
- The paper "AdaMOE: Token-Adaptive Routing with Null Experts for Mixture-of-Experts Language Models" (ArXiv: 2024-06-19), which has a very similar title, also achieves dynamic adjustment of top-k during post-training and improves accuracy while saving FLOPs during the tuning phase. Given the short time interval, the lack of comparison with this baseline can be understood, but the reviewer suggests discussing it.

**Questions:**

- The reported speedup in the paper is based on the actual inference time during evaluation tasks, which likely applies to scenarios with batch sizes of 1 or smaller. However, MoE models have large parameter volumes, and their efficiency typically becomes evident with larger batch sizes during actual deployment. The reviewer is concerned that the RL-based method of controlling top-k for each token might lead to imbalance in batch inference, potentially affecting the acceleration effect. Therefore, the reviewer would like to know the relationship between acceleration effect and batch size.
- Would combining Ada-K with the standard SFT setting yield better results? Or, under the control of Ada-K, would the performance of SFT be affected?

---

> ### Author Response · Authors · 2024-11-20
> **Rebuttal by Authors [1/2]**
>
> Dear Reviewer 3rXb,
>
> We sincerely appreciate your valuable and insightful comments. We found them extremely helpful for improving our manuscript. We will strive to address each comment in detail, one by one below.
>
> ---
>
> **W1. More Comparison**
>
> Thank you for your valuable suggestions. We fully acknowledge the two additional experimental setups you proposed. Accordingly, we have conducted supplementary experiments for the two threshold-based baselines mentioned in Table 4 (*i.e.*, MoED and D2D) under the both settings you specified, namely "LoRA ft w/o Router" and "Full ft w/o Router." The results are presented in the following table:
>
>   |   Method  | Tune Part | Trainable Parameter↓   | Acc↑ |  Rate↑ |
>   | :---------- | :------ | :-----: |:-----: |:-----: |
>   | **Default Qwen1.5-MoE-A2.7B** | |||
>   | Top-K (k = 4)| N/A | 0M  | 54.43 | 0.0% |
>   | **MoED (p = 0.3)**||||
>   | MoED  | Router | 2.95M  | 53.45 | 32.4% |
>   | MoED  | LoRA ft w/o Router | 830M | 54.06 | 31.5% |
>   | MoED | Full ft w/o Router | 14.3B  | 54.17 | 30.9% |
>   | **MoED (p = 0.4)**||||
>   | MoED | Router | 2.95M  | 53.60 | 28.6% |
>   | MoED  | LoRA ft w/o Router | 830M | 54.23 | 28.8% |
>   | MoED  | Full ft w/o Router | 14.3B  | 54.42 | 26.7% |
>   | **D2D (τ = 0.1)**||||
>   | D2D | Router | 2.95M  | 53.73 | 27.8% |
>   | D2D  | LoRA ft w/o Router | 830M | 54.58 | 27.1% |
>   | D2D  | Full ft w/o Router | 14.3B  | 54.76 | 28.2% |
>   | **D2D (τ = 0.2)**||||
>   | D2D  | Router | 2.95M  | 53.64  | 31.5% |
>   | D2D  | LoRA ft w/o Router | 830M | 54.50 | 30.9%  |
>   | D2D | Full ft w/o Router | 14.3B  | 54.55 | 32.2% |
>   | **Ours** | |||
>   | Ada-K | Allocator | 2.95M | 55.13 | 35.5% |
>
>   The results confirm that freezing the router while fine-tuning other parameters indeed enhances performance. This finding has been validated by both threshold-based baselines and various threshold settings. However, Ada-K still demonstrates a performance advantage, particularly considering that it only requires tuning less than 3M parameters.

---

> ### Author Response · Authors · 2024-11-20
> **Rebuttal by Authors [2/2]**
>
> **W2. Related Works**：
>
> Thank you for the thorough investigation. Although AdaMOE is a concurrent work, we will include it in our references and discussion.
>
> We would like to highlight the following distinctions between AdaMOE and our Ada-K:
>
> * **Technical Implementation**: Ada-K completely freezes the parameters of MoE-based LLMs, using the newly trained allocators to adaptively adjust the $k$ values for each token, thereby realizing dynamic resource allocation. AdaMOE introduces computation-free null experts and employs QLoRA to train new gates alongside the existing experts. The trained model route some tokens to null experts to reduce computational load. In summary, **Ada-K requires training much fewer parameters and does not necessitate any modifications to the original model parameters, making it overall more concise and efficient**.
>
>
> * **Allocation Principles**: Ada-K represents a more targeted  resource allocation strategy. It not only compresses the expert resources for less significant tokens (similar to the effect of null experts in AdaMOE) but also **strategically intensifies the modeling capabilities for important tokens by reallocating more resources, a feature that AdaMOE does not support**.
>
> * **Experimental Results**: Ada-K has been validated on four mainstream MoE-based LLM, achieving an average performance increase while compressing over 25% of FLOPs. In contrast, AdaMOE is tested only on Mixtral-8x7B, achieving approximately 15% FLOPs compression.
>
> **Q1. Batch Size Ablation**：
>
> Thank you for your insightful comments. We would like to clarify the following points in response:
>
> * **Inference Settings**: All inference speedup tests are conducted using 8 NVIDIA A800 GPUs, with a consistent total batch size of 16 set for all benchmarks. We utilize expert parallelism, with different experts distributed across various GPUs, where each GPU only processed a token group corresponding to the experts on that device.
>
> * **Balanced Expert Load**: Actually, the variance in expert load distribution before and after the Ada-K training is minimal, maintaining a consistent and balanced allocation. This stability is achieved by freezing the original model parameters, particularly the routers responsible for selecting the experts. This visualization is reported in Fig.6 of the original manuscript. In other words, **Ada-K uniformly and fairly reduces the computational load allocated to each expert**. This characteristic is particularly beneficial for batch inference, which we will have a discussion in the following point.
>
> * **Experiment Evaluation**: In response to your guidance, we further conduct an ablation study on inference speed across different batch sizes  based on Mixtral-8x7B, detailed in the table below:
>
>    | Batch Size | Speedup   |
>    |------------|-----------|
>    | 1          |  1.248×   |
>    | 4          |  1.267×   |
>    | 16 (default) | 1.284×  |
>    | 32         | 1.288×    |
>    | 64         | 1.285×    |
>
>   The results indicate that at smaller batch sizes, due to fewer tokens per batch, the variability in the number of tokens processed by each expert may be greater, which may introduce randomness into acceleration effect evaluations. However, as the batch size increases, the growth in token counts stabilizes the expert loads towards a uniform distribution. The advantages of Ada-K in uniformly reducing computations for each expert are more consistently demonstrated.
>
> **Q2. Ada-K + SFT**：
>
> Thank you for your interesting question.
>
> * As some of the SFT data used in the baseline models is in-house, we regret that we could not perform a completely fair comparison under the control of Ada-K with SFT data.
>
> * However, we have conducted some data ablation studies, as detailed in Table 6 of the original manuscript. When we trained the allocators using an equivalent amount of SFT data, the effects were similar to those obtained with pre-training data.

---

> > ### Author Response · Authors · 2024-11-27
> > **Official Comment by Authors**
> >
> > Dear Reviewer 3rXb,
> >
> > As the discussion deadline is approaching, we are actively looking forward to your valuable feedback and would be very grateful if you could take a moment to review our responses.
> >
> > We sincerely appreciate your precious time and consideration!

---

> > ### Public Comment · ~Zewen_Jin1 · 2025-03-19
> > **Statements about AdaMoE**
> >
> > Thanks for your great work. I have some doubts about the following issues.
> > 1. AdaMoE supports increasing the Top-K value for important tokens. The statement of "... strategically intensifies the modeling capabilities for important tokens by reallocating more resources, a feature that AdaMOE does not support." is wrong.
> > 2. The authors mention that "Although AdaMOE is a concurrent work, we will include it in our references and discussion.". However, the current version (Camera Ready Revision) still lacks the discussion of AdaMoE.

---

### Official Review · Reviewer_vr7C · 2024-11-05

**Soundness:** 2
**Presentation:** 2
**Contribution:** 3
**Rating:** 6
**Confidence:** 4

**Summary:**

The paper studies the dynamic routing strategy in MoE architectures. Conventional MoE architectures use a static Top-K routing, activating a fixed number of experts regardless of the token's complexity or importance. They propose the Ada-K routing strategy. Ada-K routing dynamically adjusts the number of activated experts based on the significance of each token, balancing efficiency and performance. The allocator module is based on RL. Ada-K achieves over 25% reduction in FLOPs and 20% faster inference speeds, with performance improvements across various benchmarks.

**Strengths:**

MoE is a scalable solution that balances parameter increase with computational cost. Targeting the limitations of prior efforts with a fixed number of experts, this paper works to make the expert number dynamic, which can bring additional efficiency and potential performance gains. The experiments include studies across multiple scales of models to test the effectiveness of the proposed method. An advantage of Ada-K is that it is pluggable, making it applicable across different MoE-based LLMs.

**Weaknesses:**

1. The Ada-k routing design works for the post-training of MoE-based LLMs. For post-training, as the gate is already trained, a question is whether it is necessary to use RL to learn the gate selection again, which complicates the overall design. For instance, a simple solution is to distill Top-k selection into binary selection with a separate gate, similar to the MoD design. The paper doesn't include comparison studies with this naive Top-k selection distillation, making it hard to say whether RL is necessary. I also didn't see a concrete motivation for using DRL.

2. Some parts of the paper writing need to be improved. For instance, for each layer $l$ equipped with the allocator, the state design and action design for the DRL agent are unclear.

**Questions:**

1. Is the usage of DRL necessary? How is the performance of the proposed method compared to naive Top-k selection distillation?

2. What is the state space of the DRL agent?

3. As each layer $l$ is paired with a DRL agent, are all DRL agents trained together or in a layer-by-layer manner? Will the selection for a layer $l$ influence the selection of layer $k$, where $k\neq l$? If so, how is the influence taken into account in the desigm?

---

> ### Author Response · Authors · 2024-11-20
> **Rebuttal by Authors [1/2]**
>
> Dear Reviewer vr7C,
>
> We sincerely appreciate your valuable and insightful comments. We found them extremely helpful for improving our manuscript. We will strive to address each comment in detail, one by one below.
>
> ---
>
> **W1 & Q1. DRL Necessity**
>
> We wish to address your concerns with the following three points:
> * **Gate Training**: We wish to kindly clarify an unintended misunderstanding: **we do not "*use RL to learn the gate selection again*"** as you mentioned in the review. Actually, throughout the Ada-K training process, the original gates (and other parameters in the baseline models) remain frozen. We only use DRL to train the newly introduced allocators. The gates, having been effectively pre-trained, **retain their original ability to determine which experts to select**, while the allocators are responsible for deciding how many experts to select. We have detailed related settings in L185-L186 of the manuscripts.
>
> * **Performance Comparison**: We greatly appreciate your suggestion to compare Ada-K with naive Top-K selection distillation. For a comprehensive analysis, we integrate a binary selection gate at three different levels, similar to the MoD design.
>   - **Expert-Level MoD**：Each expert is assigned a new binary gate. A token will then use these binary gates to decide whether to engage the corresponding expert.
>   - **MoE-Level MoD**：A new gate is introduced to decide whether each token should bypass the corresponding MoE sublayer.
>   - **Layer-Level MoD**: It is the classic MoD design, where a new gate is introduced to decide whether each token should bypass the corresponding Transformer layer (including both the Self-Attention and MoE sub-layers).
>
>   The comparison results based on Mixtral-8x7B are summarized in the table below. For fair comparison, we adopt the same training and data setting. We set the capacity, which is a hyperparameter used in MoD to decide whether to skip computations, for the three variants to ensure they have similar FLOPs to Ada-K. It enables a fair comparison of average accuracy.
>
>
>   | Method | Avg Acc↑ | Act↓ |FLOPs↓ |
>   |:--|:--:|:--:|:--:|
>   | Expert-Level MoD |  65.47 | 1.43 |4.58T |
>   | MoE-Level MoD |   64.96 | 1.39 |4.41T |
>   | Layer-Level MoD | 62.42 | 1.38 | 4.36T |
>   | Ada-K |  68.19 |  1.40 |4.42T |
>
>   The comparison highlights several key advantages of Ada-K: **(1) Performance Superiority**: Ada-K achieves significantly better performance compared to these MoD variants while maintaining similar FLOPs. **(2) Pure Dynamic Routing Decision**: Unlike binary gate methods that require pre-defined capacity thresholds, the decision-making process for Ada-K is fully learnable. This eliminates the need for manually setting thresholds for specific scenarios or models, offering significant flexibility and generalizability. **(3) More Reasonable Allocation.**: These binary selection gates, when choosing to skip computations, functions similarly to allocating fewer experts to certain tokens in Ada-K. However, the ability of Ada-K to adaptively select critical tokens and allocate them with more expert resources is something that MoD-based methods struggles to offer. This adaptability is a key reason for its superior performance.
>
> * **Necessity of DRL**: The reason why we employ RL-driven allocators is to achieve efficient and fine-grained expert resource allocation through end-to-end training. Specifically, DRL is necessary in our design for three reasons:
>   - For **efficiency**, we aim for the entire training process to be end-to-end to achieve holistic optimal learning. However, since the number of experts assigned to each token is sampled from allocator's output distribution, **this sample operation is inherently non-differentiable**, making it unrealistic to optimize directly via standard backpropagation.
>   - For **fine-grained**, we incorporate allocators at each layer, enabling it to make both token-specific and layer-specific decisions. The overlay of decisions across layers results in a dynamic and continuous decision-making process, which is highly complex.
>   - For **performance**, besides naive Top-K selection distillation you mentioned, we also compare two other threshold-based adaptive computation methods (*i.e.*, MoED and D2D) in Table 4 of the manuscript. Ada-K also demonstrates the performance advantages over them.
>
>   Considering the above, we employed PPO algorithm, known for **the robustness in complex decision-making**. In this way, the allocators are optimized through **policy gradients**, without the necessity of standard backpropagation.

---

> > ### Author Response · Authors · 2024-11-20
> > **Rebuttal by Authors [2/2]**
> >
> > **W2 & Q2. State and Action Space**
> >
> > Actually, **we have introduced the design of the allocators' state and action in lines L192-L195**. We apologize for not highlighting related points. As stated in the manuscript, the representation of a token $x$ at layer $l $ is considered the state $s_l$, and the number of activated experts $c_l$, determined through sampling, constitutes the action taken by the agent (*i.e.*, allocator). The action space traverses all possible values for the number of experts that could be activated.
> >
> > **Q3. Agent Training**
> >
> > We thank the reviewer for the insightful question. Below are our responses:
> >
> > * **Training of DRL Agents**: All DRL agents are **trained simultaneously** in an end-to-end fashion.
> >
> > * **Influence Between Layer-wise Decisions**: There is indeed an influence between decisions (*i.e.*, expert number selections) at different layers, as each layer's decision is influenced by the decisions made in previous layers. In this manner, layer-wise decisions accumulate progressively, forming a **global decision chain**. This chain is then evaluated through a global reward signal to assess the quality of the cumulative decisions. Leveraging policy gradients from DRL, layer-wise decisions are optimized globally, enabling the model to effectively coordinate them for better layer interaction and improved overall performance.

---

> > > ### Comment · Reviewer_vr7C · 2024-11-23
> > >
> > > I thank the authors for their efforts in the rebuttal and would keep my score of accepting.

---

> > > > ### Author Response · Authors · 2024-11-23
> > > > **Official Comment by Authors**
> > > >
> > > > Dear Reviewer vr7C,
> > > >
> > > > Thank you for your recognition of our work and the time and effort you have invested as a reviewer!
> > > >
> > > > We will adhere to your valuable suggestions to refine our manuscript accordingly.

---

### Author Response · Authors · 2024-11-21
**Overall Response**

We thank reviewers for all the valuable feedback, and the positive comments on meaningful research perspective (Reviewer vr7C, Reviewer 3rXb, Reviewer gGQ7, Reviewer wxU​​n, Reviewer 6jjx),  potential contributions to the community (Reviewer 6jjx, Reviewer gGQ7, Reviewer 3rXb, Reviewer vr7C), good writing (Reviewer 6jjx, Reviewer gGQ7, Reviewer wxU​​n) and abundant evaluations and ablations (Reviewer vr7C, Reviewer 3rXb, Reviewer gGQ7, Reviewer wxU​​n, Reviewer 6jjx).

We address all the reviewers' comments below and have incorporated all feedback in the revised manuscript. We sincerely aspire that our detailed rebuttal will dispel any uncertainties or misunderstandings which reviewers may have raised regarding our manuscript, thus contributing positively to the final ratings of this work. If any additional experiments are needed to further demonstrate the potential of Ada-K, we will do our utmost to supplement the relevant experiments during the valuable discussion period.

---

### Meta-Review · Area_Chair_dFR4 · 2024-12-23

**Metareview:**

This paper proposes the Ada-K routing strategy, in contrast to the existing popular "top-K" routing. Ada-K routing dynamically adjusts the number of activated experts based on the significance of each token, balancing efficiency and performance. The allocator module is trained via PPO to work around the non-differentiability. Ada-K achieves over 25% reduction in FLOPs and 20% faster inference speeds, with performance improvements across various benchmarks.

The idea intuitively makes sense. The authors did good execution to show the method's effectiveness.

**Additional Comments On Reviewer Discussion:**

The authors provided a comprehensive rebuttal to address reviewers' concerns. After rebuttal, all reviewers stay positive about this paper.

---

### Decision · Program_Chairs · 2025-01-22

Accept (Poster)